# Analysis of Transcriptome, Selected Intracellular Signaling Pathways, Proliferation and Apoptosis of LNCaP Cells Exposed to High Leptin Concentrations

**DOI:** 10.3390/ijms20215412

**Published:** 2019-10-30

**Authors:** Marta Szyszka, Lukasz Paschke, Marianna Tyczewska, Karol Jopek, Piotr Celichowski, Paulina Milecka, Gulnara Sultanova, Ewelina Stelcer, Agnieszka Malinska, Ludwik K. Malendowicz, Marcin Rucinski

**Affiliations:** 1Department of Histology and Embryology, Poznan University of Medical Sciences, Swiecickiego 6 Street, 60-781 Poznan, Poland; mszyszka@ump.edu.pl (M.S.); paschkelukasz@gmail.com (L.P.); maritycz@ump.edu.pl (M.T.); kjopek@ump.edu.pl (K.J.); pcelichowski@ump.edu.pl (P.C.); paulina.a.grabowska@gmail.com (P.M.); ewelina.stelcer@wco.pl (E.S.); amalinsk@ump.edu.pl (A.M.);; 2West Kazakhstan Marat Ospanov Medical University, Maresyev 68 Street, Aktobe 030019, Kazakhstan; stomfak.zkgmu@mail.ru; 3Radiobiology Lab, Greater Poland Cancer Centre, Garbary 15th Street, 61-866 Poznan, Poland; 4Department of Electroradiology, Poznan University of Medical Sciences, Garbary 15th, 61-866 Poznan, Poland

**Keywords:** prostate cancer cells, LNCaP, leptin, transcriptome analysis, proliferation, apoptosis, intracellular signaling pathways

## Abstract

Leptin, the first discovered adipokine, has been connected to various physiological and pathophysiological processes, including cancerogenesis. Increasing evidence confirms its influence on prostate cancer cells. However, studies on the effects of leptin on the proliferation and apoptosis of the androgen-sensitive LNCaP line of prostate cancer cells brought conflicting results. Therefore, we performed studies on the effects of high LEP concentration (1 × 10^−6^ M) on gene expression profile, change of selected signaling pathways, proliferation and apoptosis of LNCaP cells. RTCA (real-time cell analyzer) revealed inhibitory effect of LEP on cell proliferation, but lower LEP concentrations (10^−8^ and 10^−10^ M) did not affect cell division. Moreover, flow cytometry with a specific antibody for Cleaved PARP-1, an apoptosis marker, confirmed the activation of apoptosis in leptin-exposed LNCaP line of prostate cancer cells. Within 24 h LEP (10^−6^ M) increases expression of 297 genes and decreases expression of 119 genes. Differentially expressed genes (DEGs) were subjected to functional annotation and clusterization using the DAVID bioinformatics tools. Most ontological groups are associated with proliferation and apoptosis (seven groups), immune response (six) and extracellular matrix (two). These results were confirmed by the Gene Set Enrichment Analysis (GSEA). The leptin’s effect on apoptosis stimulation was also confirmed using Pathview library. These results were also confirmed by qPCR method. The results of Western Blot analysis (exposure to LEP 10 min, 1, 2, 4 and 24 h) suggest (after 24 h) decrease of p38 MAPK, p44-42 mitogen-activated protein kinase and Bcl-2 phosphorylated at threonine 56. Moreover, exposure of LNCaP cells to LEP significantly stimulates the secretion of matrix metallopeptidase 7 (MMP7). Obtained results suggest activation of apoptotic processes in LNCaP cells cultured at high LEP concentration. At the same time, this activation is accompanied by inhibition of proliferation of the tested cells.

## 1. Introduction

Leptin (LEP) is predominantly produced and secreted by adipose tissue, functioning mainly in the regulation of energy balance and food intake. Due to the fact that LEP and its receptors are widely expressed, LEP is also a multifunctional pleiotropic hormone, acting as an auto, para and endocrine signal. The increasing body of evidence indicates that the influence of LEP extends to several hypothalamic-pituitary-endocrine axes including adrenal, thyroid, pancreatic islands. Moreover, the role of leptin in immune function, haematopoiesis, osteogenesis and angiogenesis was also documented [1]. LEP levels in human blood (LEP normal blood levels are reported to be 1–15 ng/mL) increase in several diseases, including metabolic disorders leading to obesity. Hyperleptinemia is also associated with the pathogenesis of some cancer types. [2,3,4,5,6,7]. It was shown that elevated serum and tissue LEP level is involved in the pathogenesis of lung cancer and tumor metastasis [8]. Hyperleptinemia is also associated with metastases of melanoma to lymph nodes [9] and is considered as pathophysiological factor in pathogenesis of breast cancer [10]. In the literature, there are also several reports about the role of LEP in the progression of prostate cancer [11,12,13,14,15,16].

Earlier studies on rats identified various isoforms of the LEP receptor in the rat prostate and seminal vesicles and suggested that this cytokine exerts a stimulating effect on the proliferation of epithelial cells of these organs [17,18,19,20]. However, data on LEP receptor expression in human prostate are very scarce. Cioffi et al. identified different variants of the LEP receptor (LEPR) using the Northern blot method in human prostate tissue [21]. In other studies, using RT-PCR method, the following LEPR variants were expressed in tissue samples from benign prostatic hyperplasia (BPH): variants 4 and 2 (in all five samples studied), var. 5 (3/5) and var. 6 and 3 (4/5) [22]. Moreover, Leze et al. demonstrated that incubation of human hyperplastic prostate tissue fragments with LEP (50 ng/mL) for 3 h significantly stimulates proliferation of epithelial cells and expression of pro-apoptotic *BAX* gene [23].

Difficulties in acquiring appropriate prostate fragments have led various research groups to perform research on various human normal prostate and prostate cancer cell lines. However, the expression of different variants of LEPR in these cells differs significantly [24]. There are also differences in the results of research on the role of LEP in the regulation of proliferation and apoptosis of these cell lines. In the case of LNCaP cells LEP either does not change the proliferation rate of these cells [24,25,26], may stimulate it [27,28], and at high concentrations of the cytokine tested (1 × 10^−6^ M) may inhibit the growth of these cells [24]. Taking into account the latter finding, it should be stressed that in comparable concentrations of LEP (12.5 µg/mL) no proliferation changes were observed in LNCaP cells [25], whereas in DU145 cells this concentration of cytokine stimulated proliferation of studied cells [29].

Considering the abovementioned discrepancies, we decided to analyze the effect of high concentrations of LEP on proliferation, gene expression profile and changes in selected signaling pathways of LNCaP cells.

## 2. Results

### 2.1. Leptin at a Dose of 1 × 10^−6^ M Exerted an Inhibitory Effect on Proliferative Activity of LNCaP Cells and Stimulate Apoptosis

Using a real-time proliferation assay, we examined the effect of LEP, at concentrations of 10^−6^, 10^−8^, and 10^−10^ M, on the proliferation rate of LNCaP cells. As we shown in Figure 1A, LEP at a dose of 1 × 10^−6^ M leads to a significant inhibition of LNCaP cells proliferation. Both lower LEP concentrations (10^−8^, 10^−10^ M) did not affect the proliferation rate of cultured cells. Therefore, further studies were performed with LEP at a dose of 1 × 10^−6^ M in relation to the control group. Based on median fluorescence intensity, LNCaP cells treated with the highest LEP concentration (1 × 10^−6^ M) revealed 30% higher level of apoptosis in comparison with untreated cells (control) (Figure 1B). In LNCaP cells treated with lower concentrations of leptin (10^−8^ and 10^−10^ M) we did not observed statistically significantly differences (data not shown).

### 2.2. Leptin at a Dose of 1 × 10^−6^ M Significantly Modulates the Transcriptomic Profile of LNCaP Cells

The GeneChip Human Genome U219 Array Strips used in the current study allowed the simultaneous examination of the gene expression of 19,285 human transcripts. The transcriptome study was performed 24 h after LEP administration (1 × 10^−6^ M) to the culture medium. The transcriptome profile was compared with the untreated (control) group. The general profile of transcriptome changes was shown as a volcano plot (Figure 2A).

We assumed the following selection criteria for differentially expressed genes (DEG): an expression fold difference > absolute 2 and an adjusted *p*-value ≤ 0.05. According to these criteria, 297 genes were up-regulated, while 119 were down-regulated as consequence of LEP action. The ten genes with the highest and lowest fold change values are presented in tabular format displaying the gene symbol, gene name, fold change and adjusted *p*-value (Figure 2B). These genes were characterised by high fold change values, especially for up-regulated genes (fold range for up-regulated genes: 99.66–17.64, and for down-regulated genes: −15.70–−3.63). Among others these genes include: chemokine (C-C motif) ligand 20 (CCL20, fold = 99.66), matrix metallopeptidase 7 (MMP7, fold = 62.57), tumor necrosis factor, alpha-induced protein (TNFAIP3, fold = 23.29).

### 2.3. LEP at 1 × 10^−6^ M Concentration Exerts a Significant Effect on the Genes Involved in the Regulation of the Following Biological Processes: Apoptosis, Immunological Response and Extracellular Matrix Organisation

To determine which biological processes can be regulated by LEP, we performed an analysis of the enrichment in the relevant ontological groups from the GO BP Direct database. A whole set of differentially expressed genes (DEGs) consisting of 416 genes (297 up- and 119 down-regulated) was subjected to functional annotation and clusterization using the Database for Annotation, Visualization, and Integrated Discovery (DAVID) bioinformatics tools. The result of this analysis was shown as a bubble plot (Figure 3) where we presented only ontological groups fulfilling the following criteria: adjusted *p*-values below 0.05 and minimal number of genes per group >10. 

The performed analysis revealed 21 ontological groups. The majority of them involved: regulation of proliferation and apoptosis (n = 7), imunological response (n = 6) and organization of extracellular matrix (n = 2), signaling of MAPK (n = 1). The detailed characteristics of the above mentioned ontological groups were as follows: “GO:0071356~cellular response to tumor necrosis factor” (n = 11, adj. *p* value = 0.02), “GO:0043065~positive regulation of apoptotic process” (n = 16, adj. *p* value = 0.04), “GO:0042981~regulation of apoptotic process” (n = 14, adj. *p* value = 0.03), “GO:0042127~regulation of cell proliferation” (n = 12, adj. *p* value = 0.03), “GO:0033209~tumor necrosis factor-mediated signaling pathway” (n = 11, adj. *p* value = 0.02), “GO:0008285~negative regulation of cell proliferation” (n = 20, adj. *p* value = 0.026), “GO:0006915~apoptotic process” (n = 35, adj. *p* value = 1.30 × 10^−4^), “GO:0060337~type I interferon signaling pathway” (n = 11, adj. *p* value = 5.03 × 10^−4^, “GO:0060333~interferon-gamma-mediated signaling pathway” (n = 13, adj. *p* value = 6.41 × 10^−5^), “GO:0050852~T cell receptor signaling pathway” (n = 13, adj. *p* value = 0.02), “GO:0045087~innate immune response” (n = 24, adj. *p* value = 0.02), “GO:0006955~immune response” (n = 28, adj. *p* value = 5.32 × 10^−4^), “GO:0006954~inflammatory response” (n = 29, adj. *p* value = 9.12 × 10^−5^), “GO:0030198~extracellular matrix organization” (n = 17, adj. *p* value = 2.03 × 10^−3^), “GO:0022617~extracellular matrix disassembly” (n = 11, adj. *p* value = 1.94 × 10^−4^), “GO:0000165~MAPK cascade” (n = 16, adj. *p* value = 0.03).

The above results were confirmed by another powerful bioinformatic tool—Gene Set Enrichment Analysis (GSEA). In this analysis, the fold change values of all genes were log2 transformed and ranked according their logFC. Afterwards, these values were used for a 1000 times permutation test to calculate the enrichment score (ES) within predefined gene sets from the Hallmark database. Enrichment scores were normalized regarding gene set size and presented as normalised enrichment score (NES). The result of the GSEA analysis for the ranked log2 fold change values of LEP (1 × 10^−6^ M) vs. control group was presented in Figure 4. The ten hallmark database groups with the highest absolute NES value were presented in Figure 4A. Within the gene ranks column each vertical line represents one gene and its position depends on the logFC value (around 0 are genes with high logFC value on which LEP had a stimulating effect—enriched genes, on the right side around 19,285—there are genes whose expression due to LEP action is lowered and had a low logFC value—depleted genes). Despite a different methodological approach, the GSEA analysis presents relatively similar groups as shown in the analysis of ontological groups by DAVID. This groups concern: apoptosis/proliferation and immunological processes. The analysis regarding the position of single genes on gene rank scale and consequently the NES values of individual groups indicates that LEP stimulates expression of genes in the immune response hallmark group. Genes involved in the regulation of apoptosis are also strongly stimulated by LEP. The analysis of enrichment within the Hallmark apoptosis group is characterized by a high positive value of NES = 4.25 and includes genes with a very high logFC value as it is shown in Figure 4C. Interestingly, the GSEA analysis also showed a significant decrease in expression of genes closely related to proliferation belonging to the following sets of hallmark database “mitotic spindle” (NES = −2.92) and “G2M checkpoint”. (NES = −4.79). These groups are composed of the genes with very low logFC values and therefore their expression is suppressed by LEP (Figure 4C).

### 2.4. Detailed Analysis of the LEP-Regulated Gene Expression in LNCaP Cells

Because DAVID and GSEA analyses indicated potent regulation by LEP of genes related to proliferation/apoptosis, as well as due to a significant decrease in proliferation under the influence of LEP in RTCA study, in the next step we analysed specific genes belonging to the following ontological groups: “apoptotic process”, “regulation of cell proliferation”, “regulation of apoptotic process”, “negative regulation of cell proliferation”, “positive regulation of apoptotic process”. The fold change values of genes forming the mentioned groups were used to calculate Z-score which indicates whether the process decreased (negative value) or increased (positive value) and presented as circular scatter plots. Z-score was calculated automatically using the GOplot library. Despite the presence of several expression decreased genes, all of the analysed ontological groups were characterised by a positive value of Z-score, confirming the stimulating effect exerted by LEP on the given processes. (Figure 5A). Due to ambiguous nature of the gene ontology structure, single genes can often be assigned to many ontological terms. For this reason, the relationship between genes and GO terms were mapped with circos plots with visualization of logFC values and gene symbols (Figure 5B). The strongest up-regulated genes from the examined ontological groups included, among others: *BIRC3*—baculoviral IAP repeat containing 3, *FAS*—Fas cell surface death receptor, *TNFAIP3*—tumor necrosis factor, alpha-induced protein 3, *TNF*—tumor necrosis factor, *GADD45G*—growth arrest and DNA-damage-inducible protein GADD45 gamma.

The leptin’s effect on apoptosis stimulation was also confirmed using Pathview library. Fold values of DEGs were mapped with appropriate colours for each gene forming “Apoptosis” (Figure 6) and “p53 signaling pathway” (Figure 7). Green colour indicates statistically significant gene up-regulation and red colour refers to down-regulated genes. Grey colour marked genes whose expression was not significantly changed. Analogous to the David analysis with circosplot visualization, most of the genes displayed in “Apoptosis” and “p53 signaling pathway” were up-regulated. This result confirms the stimulatory effect of LEP on apoptosis activation, moreover, it is consistent with the previously described LEP effect causing a decrease in LNCaP cells proliferation.

The expression of several differentially expressed genes was also validated using qPCR. Significantly different genes were selected for validation according to the Volcano Plot and the list of genes with the largest fold changes obtained from the datasets with differentially expressed genes (Figure 2A,B). Our findings confirmed the effect exerted by LEP on the expression of examined genes, namely: *BMX*—BMX non-receptor tyrosine kinase, *C11orf92*—chromosome 11 open reading frame 92, *KLK4*—kallikrein-related peptidase 4, *MYLK*—myosin light chain kinase, RIMS1—regulating synaptic membrane exocytosis 1, *BIRC3*—baculoviral IAP repeat containing, FAS—apoptosis signal receptor/cell surface death receptor, *MMP7*—matrix metallopeptidase 7, *TNFAIP3*—tumor necrosis factor, alpha-induced protein 3. In accordance with the results from microarrays, the expression of all the above-mentioned genes was stimulated by LEP at 1 × 10^−6^ M concentration (Figure 8).

### 2.5. LEP Regulates Several Key Factors of Signaling Pathways Involved in Apoptosis, Proliferation and Migration

In the next step, we studied the contribution of LEP in modulation of several important signaling pathways involved in apoptosis/proliferation regulation. The study was carried out through the incubation of LNCaP cells with LEP in the following time series: 0 min (control), 10 min, 1 h, 2 h, 4 h and 24 h. Obtained results were compared to the control group. Activation of signaling pathways was analyzed using antibodies directed to p38 mitogen-activated protein kinase, p44-42 mitogen-activated protein kinase, Bcl-2 phosphorylated at threonine 56 and total Stat1. Densitometric analysis was normalized in relation to GAPDH levels and presented in Figure 9. Up to the 2nd hour of culture in the presence of LEP we did not find its effect on p38 mitogen-activated protein kinase, while from the 4th to the 24th hour we observed a significant decrease in p38 MAPK activation. During the studied time interval, p44-42 mitogen-activated protein kinase was consistently decreased and achieved a statistically significant reduction by the 24th h. Bcl-2 phosphorylated at threonine 56 increased rapidly at 10 min of incubation. This increase was maintained until the 2nd hour of culture. From the fourth hour, it was significantly lowered and this reduction was also maintained at the 24th hour. Stat1 expression was relatively stable up to the 2nd hour and then gradually increased to reach a statistically significant value at 24 h of culture.

Matrix metallopeptidase 7—MMP7, was one of the genes whose expression was strongly stimulated by LEP, therefore, we decided to check whether the increase of its expression was also reflected in the protein level. For this purpose, the level of the secreted MMP7 was determined in cell culture medium using ELISA method. As we have shown in Figure 10, LEP significantly stimulates the secretion of MMP7, that was in line with the previously described microarray data (Figure 2A,B) and real-time qPCR validation (Figure 8).

### 2.6. Leptin (LEP), Leptin Receptor (LEPR) and Its Main Downstream Signaling Genes (JAK2, STAT3) Are Downregulated in Prostate Adrenocarcinoma

Analysis of RNAseq data for 52 normal prostate (control) and 498 prostate adenocarcinoma obtained from TCGA database, revealed that LEP, LEPR, JAK2 and STAT3 are statistically significant downregulated in prostate tumors in relation to normal (control) prostate (Figure 11).

## 3. Discussion

It is generally accepted that LEP affects tumor cell invasion and progression [30,31]. LEP exerts its physiological effect by binding to set of LEP receptors, extended class I cytokine receptor family, composed of six isoforms [32]. Molecular mechanism of LEP action is well described. LEP binding to all short LEP receptors (LEPR var 2–6) leads to activation of Janus-activated kinase (JAK2) with subsequent phosphorylation of insulin receptor substrates (IRS), initiating activation of the phosphoinositide 3 kinase (PI3K)/Akt pathway [29]. Activation of IRS causes also stimulation of NF-kB signaling pathways involved in cell migration and inflammatory response [33]. The long form of the receptor (LEPR var 1) contains an additional intracellular carboxy terminal motif necessary to activate STAT3 and STAT5 [34].

Recently, using multiple sets of specific primers, we have shown that only the second isoform of the LEP receptor is expressed in the LNCaP cell line [24]. Therefore, it seems that the response of LNCaP cells to LEP is mediated via JAK2 kinase activation of LEPR var 2. Moreover, the results of other studies show that STAT3 activation under the influence of LEP in LNCaP cells may occur, however only in cells transiently co-transfected with LEPR var 1 [35]. With regard to this, Deo et al. (2008) revealed that STAT3 of LEP-exposed LNCaP cells undergoes fast dose-depended phosphorylation. This finding indirectly confirms the presence of an active LEPR form in LNCaP cells [27].

The molecular mechanism exerted by LEP was determined using high-throughput microarray, real time PCR, Western Blot and ELISA methods. The use of U219 Array Strips allowed simultaneous examination of transcriptome profile by measurement of 19,285 human genes. In the present study, we identified 416 LEP-responsive genes, most of which were up-regulated (297 genes). The analysis of GO terms including DEGs, revealed that LEP participates in the regulation of apoptosis, immunological response and extracellular matrix organisation in LNCaP cells. Interestingly, a significant number of genes with the highest expression level is related to immunological responses.

Currently, no available data indicates role of LEP in the regulation of immunological processes in prostate LNCaP cell line, however several arguments support our findings. Both molecular structure of LEP and its receptor allow to include them to the cytokine family. The secondary structure of LEP is similar to the long-chain helical cytokines family, including interleukin 6 (IL-6), IL-11, CNTF and LIF. LEP receptor amino acid sequence shares a strong homology with gp-103 signal-transducing subunit of the IL-6-type cytokine receptors [36]. It was also shown that exogenous LEP causes up-regulation of pro-inflammatory cytokines in macrophages and lymphocytes T [37,38]. There are also reports indicating that the LNCaP cell line is responsive to TNF-alpha, a known pro-inflammatory signal [39]. By means of transcriptome analysis authors revealed the stimulation of pro-inflammatory processes in these cells. In line with our results, this team observed inhibition of proliferation in LNCaP cell exposed to TNF-alpha, and the associated gene expression changes are similar to those observed in our experiments.

Interestingly, the results of our study have also shown that LEP significantly stimulates expression of NF-κB family members genes: nuclear factor of kappa light polypeptide gene enhancer in B-cells 1 (*NFKB1*, fold = 3.22, *p* = 0.0001), nuclear factor of kappa light polypeptide gene enhancer in B-cells 2 (*NFKB2*, fold = 1.6, *p* = 0.005), v-rel reticuloendotheliosis viral oncogene homolog B (*RELB*, fold = 3.33, *p* = 0.0005), v-rel reticuloendotheliosis viral oncogene homolog (*REL*, fold = 1.57, *p* = 0.01). Nuclear factor kappa B (NF-κB) belongs to essential class of transcriptional regulators. NF-κB plays an important role in regulation of multiple physiological processes, among others: the immune response, migration and invasion of cancer cells [33,40,41]. Jin et al. (2008) demonstrated that neuropeptides secreted from prostate neuroendocrine cells may activate the NF-κB pathway in LNCaP prostate cell line [40]. In Du145, PC3 and LNCaP cell lines, LEP induces cell migration through NF-κB [33]. The results of our research seem to confirm the contribution of LEP in the regulation of NF-κB pathway in LNCaP cells, modulating inflammatory processes and cell migration through this signaling pathway.

In the group of genes related to pro-inflammatory response, the highest differences in expression after LEP treatment, was observed in the *CCL20* gene. C-C motif chemokine ligand 20 (CCL20) belongs to the subfamily of small chemokine C-C genes involved in inflammatory process. CCL20 is also overexpressed in many types of cancers, however its role in tumors is not fully explained. In the context of prostate cell lines, Beider et al. showed that basal mRNA of CCL20 was detectable in PC3, Du154 cells and presented at very low level in LNCaP cells [42]. Moreover, CCL20 expression in the LNCaP line is stimulated by interleukin-17A [43].

Our results also revealed a significant contribution of LEP to the regulation of extracellular matrix organization. Matrix metalloproteinase 7 (MMP7) plays essential role in prostate cancer cell invasion and epithelial to mesenchymal transition by breaking down extracellular matrix of tumor cells [44]. In this context Zhang et al. (2016), showed that overexpression of MMP7 in LNCaP cells leads to disruption of E-cadherin/β-catenin complex and releases β-catenin, thus enhancing EMT and tumor cell invasion [44]. It is astonishing that MMP7 is not expressed in a normal prostate, whereas is overexpressed in human prostate cancer [45,46]. Relatively high expression of MMP7 was observed in our LNCaP cells. MMP7 expression at mRNA and protein level, measured as secretion to cell culture medium, was significantly stimulated by LEP. MMP7 may also play role in apoptosis induction. MMP7 leads to release of membrane bound FASL that induces apoptosis of neighboring cells via death receptor FAS [47,48,49]. 

Lipocalin 2 (Lcn2) is described as ligand for matrix metallorproteinase 9, also known as neutrophil gelatinase-associated lipocalin (NGAL). Lcn2 is upregulated in several types of cancers, and has been shown to facilitate tumor progression [50]. In LNCaP cells, Lcn2 upregulation undergoes via NF- kB-dependent manner [51]. In relation to the prostate cancer cell lines there are data proving the contribution of Lcn2 in prostate cell proliferation, however, this effect is dependent on the prostate cell type. Tung et al. (2013) found that LCN2 knock-down in PC3 and DU145, reduced cell growth, by induction of cell-cycle arrest at G0/G1 phase and Lcn2 overexpression stimulates growth, migration and invasiveness of 22Rv cells [50].

Increasing evidence suggests that LEP exerts a significant effect on the proliferation and apoptosis rates of different prostate cell lines, but the results of individual studies are inconclusive. The analysis of the collected data presented in Table 1 indicates that the effect of leptin on proliferation and apoptosis of human normal prostate and prostate cancer cell lines depends on the cell line tested, the time of exposure to LEP and the concentration of this cytokine in the culture medium.

In our study, strong LEP-dependent regulation of apoptotic processes in LNCaP prostate cell line was demonstrated. This pro-apoptotic effect was triggered by LEP at the highest of the tested doses (1 × 10^−6^ M), that affected on expression changes of many apoptotic genes. It is well known that apoptotic process is divided into two stages: induction and execution. Induction of apoptosis is a multifactorial stage of apoptosis, that includes involvement of many different factors, like receptors, ligands, intracellular peptides. Presented here microarray analysis using KEGG “apoptosis” pathway showed that after 48 h exposure, among 41 DEGs involved in apoptosis most of them—39, was up-regulated in LNCaP cells. For example, the expression of *FAS* gene, which encodes a cell surface death receptor that play key role in induction of apoptosis by binding FAS ligand (FASLG) in cell membrane, was almost 3 times higher (fold = 2.95, *p* = 0.003), compared to control. This statistically significant increase was also confirmed using qPCR method. Unlike our results, Tanaka and coworkers (2008) demonstrated inhibitory effect of LEP on Fas-dependent apoptosis [55]. This result was obtained using LEP at physiological concentration, indicating the importance of the dose of the peptide administered in the experiments. On the other hand, current studies have shown that another ligand of death receptors involved in induction of apoptosis (mentioned above) —member of TNF superfamily—TNF-alpha, was also found strongly up-regulated after LEP 1 × 10^−6^ M administration (fold = 4.27, *p* = 0.0001). Moreover, the expression of *TRADD* gene that encode tumor necrosis factor receptor type 1-associated DEATH domain protein (TRADD) involved in transduction of the TNF-alpha signal downstream, was also found to be increased (fold = 1.47, *p* = 0.01). Similar changes were found in the expression of *TP53* gene (p53 protein) (fold = 1.66, *p* = 0.003). As commonly known, p53 protein plays significant role in regulation of progression through apoptosis when DNA damage is irreparable. It activates pro-apoptotic BH family peptides proteins, like BAX, BID and transcription factors like NOXA, PUMA and direct the cell to death. Here the expression level of those molecules were found to be increased (BAX fold = 1.31, *p* = 0.048; BID fold = 1.43, *p* = 0.021).

Activation of death domain receptors leads to activation of initiating caspases 8 and 10, which next, on the one hand, activates pro-apoptotic protein BID and on the other hand, activates procaspases 3 and 7—inactive forms of effector caspases 3 and 7 (CASP3, CASP7). This is followed by avalanche activation of target proteins, leading to cell death. Active protein BID activates other pro-apoptotic BH family peptides—BAX and BAK, which lead to the release of cytochrome C from the mitochondrial matrix and form apoptosome complex, while active caspase 3 cleavage the DNA repair enzyme (PARP) into active form. In our study, the expression level of several mentioned above genes involved in progression of apoptosis, were clearly elevated (*CASP8* fold = 1.81, *p* = 0.002; *CASP10* fold = 1.42, *p* = 0.019; *CASP7* fold = 2.53, *p* = 0.001). In the literature there is lot of data regarding involvement of LEP in the regulation of apoptotic processes in prostatic cells. Most of them indicated inhibitory effect of LEP on this process and promotion of cellular proliferation [20,28,52,54,56,57]. However, some studies demonstrated LEP-dependent activation of apoptotic processes. Regarding this Samuel-Mendelsohn et al. (2011) reported LEP-induced activation of apoptotic effector molecules (CASP3 and PARP) in LNCaP cells [35]. Using Western-blot analysis followed by densitometry quantification they noted dose-dependent increase in CASP3 and PARP level observed between 6 and 24-h of LEP administration (1 ng/mL of LEP). This result is in line with the results presented here. Despite the fact that we didn’t notice increase of expression of CASP3 mRNA, we detected increased expression of PARP mRNA in LNCaP cells (described on the signal path as PARP, mapped as PARP4 fold = 1.33, *p* = 0.038). Other studies of Kim and coworkers (2003) demonstrated LEP-dependent activation of caspase 3 and caspase 9 in osteoblast-lineage of primary human bone marrow stromal cells [57]. This caused increase of cytochrome c release from mitochondria and could confirm our finding on LEP-activation of apoptosis, however we didn’t note any changes in expression level of both caspases mRNA [57]. Although most genes belonging to the ontological group “apoptotic process” were up-regulated, as well as enrichment analysis of ontological groups indicated a significant stimulation of this process, several genes involved in the regulation of apoptosis are lowered under the influence of LEP, e.g., *PEG3*, *DAPK1*, *ZBTB16*, *STK3* and *AR*. It is well described that proliferation and growth of LNCaP cells is androgen dependent, where AR plays proliferative or/and apoptotic role by interaction with MAPK, EGFR, IGF, TGF beta, FGF or VEGF [58,59,60,61]. For this reason, we cannot exclude an indirect involvement of the androgen receptor in the observed LEP effect. However, this aspect requires further research.

We are aware that the research was carried out with the use of high LEP concentrations in the culture medium. However, similar LEP concentrations are also used in other in vitro experiments [62,63,64,65,66]. For example, such high LEP concentrations are used in studies related to the involvement of LEP in the regulation of pituitary gland hormone secretion or cell proliferation. In concentrations comparable to our own LEP inhibits pituitary cell proliferation in human and rat pituitary cell lines in vitro [61], stimulates FSH and LH release from pituitary cells of male and female rainbow trout [64] and TSH secretion from ovine pituitary cells [63]. As mentioned above, LEP concentration used is far from the physiological. Therefore, the use of high LEP doses in systemic administration seems to be a limiting factor, due to possible side effects. However, the potential use of high doses of LEP administered directly to the prostate should be taken into consideration. It should also be noted that LEP and its receptor are expressed within the prostate, therefore its local para and autocrine activity are not excluded, where in the intercellular areas the leptin level may be higher than in the serum. Additionally, there are numerous depots of adipose cells within the prostate that may constitute a source of locally acting leptin. This suggestion is reinforced by the fact that the expression of LEP, leptin receptor (LEPR) and its main downstream signaling genes (*JAK2*, *STAT3*) is reduced in prostate adenocarcinoma (Figure 11), suggesting that this system is involved in the mechanism of apoptosis defense in proliferating tumor cells. However, this suggestion requires further studies.

## 4. Materials and Methods

### 4.1. Prostate Cancer Cell Line

LNCaP—the human prostate carcinoma cell line (LNCaP clone FGC (ATCC^®^ CRL-1740DTM)) was purchased from ATCC (American Type Culture Collection, Manassas, Rockville, MD, USA). LNCaP cells were cultivated in the RPMI Medium 1640 (1×), supplemented with GlutaMAX, HEPES (all from Gibco, Life technologies, Carlsbad, CA, USA) and antibiotics (Antibiotic/antimycotic Sigma Aldrich, Saint Louis, MO, USA). Cells were grown in the 75 cm^2^ flask (NUNC EasyFlask with Nunclon surface, Thermo Fisher Scientific) at 37 °C in a humidified atmosphere of 5% CO_2_. The culture medium was changed every 2 days [67]. When cells reached approximately 80% confluence (about 7–8 days of cultivation), they were subcultured into 6-well plates (Nunc, Thermo Fisher Scientific, approximately 343,000 cells per one well—9.6 cm^2^), to determine the effect of LEP on LNCaP cells proliferation at mRNA/protein level. Simultaneously, cells were seeded on E-Plate 48 (Roche Applied Science, GmbH, Penzberg, Germany or ACEA Biosciences Inc., San Diego, CA, USA, approximately 12,500 cells per one well—0.3 cm^2^) to perform real time proliferation assay [24]. The applied experimental protocol was as follows: during the first 48 h of cultivation, the cells grew in a standard medium mentioned above. For the next 24 h, the cells were grown in starvation medium (FBS free). Afterwards, the cells were cultivated for 48 h, in a starvation medium supplemented with LEP (Recombinant Human Leptin, PeproTech, Germany) at the following concentrations: 0 (control), 1 × 10^−6^; 1 × 10^−8^ and 1 × 10^−10^ M. After the mentioned period, the medium and cell supernatants were collected and stored at −80 °C for further analyses. 

### 4.2. Real-Time Proliferation Assay

To verify proliferation rate of LNCaP cells we used an electrical impedance based approach—the Real-Time Cell Analyser (RTCA, Roche Applied Science, GmbH, Penzberg, Germany). The system detects variations in electrical impedance across incorporated sensor electrode arrays placed on the bottom of 16-well chamber slide plates (E-plate 16) on which the cells are seeded. Electrical impedance is measured throughout the cultivation period at a 15-minutes frequency. The main read-out of the RTCA is a dimensionless parameter named “Cell Index” which represents a relative change in electrical impedance, depending on the proliferation or apoptosis rate of the cultured cells. LNCaP cells were cultivated in the same groups and experimental layout as described above. Each group was seeded in the eight wells of E-plates in a final volume of 200 µL per well. Cell index was normalised (normalised cell index) at the time point of LEP administration, using software provided by the manufacturer (RTCA Software, Version 1.2, November 2009). LNCaP cells were cultivated with LEP till reaching the plateau phase in the control group (total time: 196 h). Each experiment was repeated at least three times.

### 4.3. Flow Cytometry Analysis of Cleaved PARP-1

LNCaP cells (un- and treated with different concentrations of leptins) were stained for cPARP with the PE Mouse Anti-Cleaved PARP (Asp214) antibody (562253, BD Biosciences, NJ, USA) according to manufacturer’s instructions. Briefly, 1 × 10^6^ un- and treated cells were fixed and permeabilized with BD Cytofix/Cytoperm Fixation/Permeabilization Solution for 30 min at room temperature. Then, the additional permeabilization and fixation was performed. The fixed cells were washed with BD Perm/Wash Buffer and stained with appropriate antibody (5 μL/test) for 20 min at room temperature. The stained and washed cells were resuspended in 500 μL PBS and analyzed with a flow cytometer (CytoFLEX, Beckman Coulter, CA, USA). Fluorescence intensity in arbitrary units was plotted in histograms and the mean fluorescence intensity was calculated. Data were analyzed using FlowJo software (FlowJo v10; LLC, Ashland, OR, USA).

### 4.4. RNA Isolation

The applied methods were described earlier [68]. The total RNA was extracted using TRI Reagent (Sigma-Aldrich, St. Louis, MO, USA) then purified on columns (NucleoSpin Total RNA Isolation, Qiagen GmbH, Hilden, Germany). The amount of total RNA was determined by optical density at 260 nm and its purity was estimated by 260/280 nm absorption ratio (higher than 1.8) (NanoDrop spectrophotometer, Thermo Scientific, Waltham, MA, USA). The RNA integrity and quality were checked in a Bioanalyser 2100 (Agilent Technologies, Inc., Santa Clara, CA, USA). The resulting RNA integrity numbers (RINs) were between 8.5 and 10 with an average of 9.2. Each sample was diluted to the RNA concentration of 100 ng/μL, at the OD260/OD280 ratio of 1.8/2.0. From each RNA sample, 100 ng of RNA was taken for microarray experiments. The remaining amount of isolated RNA was used for RT-qPCR study.

### 4.5. Reverse Transcription

Reverse transcription was performed using Transcriptor High Fidelity Reverse Transcriptase enzyme blend for high fidelity two-step RT-PCR of RNA (Roche, Basel, Switzerland) with Oligo(dT) as primers at a temperature of 45 °C for 40 min (Thermocycler UNO II, Biometra, Göttingen, Germany). For a single reaction, 1 μg of total RNA was used. The RT was carried out in standard final volumes (20 μL). After RT each cDNA containing sample was diluted with 100 µL of RNase-free water.

### 4.6. Q-PCR

Q-PCR was performed using the Lightcycler 2.0 instrument (Roche, Basel, Switzerland) with the 4.05 software version. SYBR green detection system was applied as described earlier [67]. Every of 20 µL reaction mixtures contained 2 µL template cDNA (standard or control), 0.5 µM of specific primer and a previously determined optimum MgCl_2_ concentration (3.5 µM for one reaction). Light Cycler Fast Start DNA Master SYBR Green I mix (Roche) was used. The real-time PCR program included 10 min denaturation step to activate the Taq DNA Polymerase, followed by a three-step amplification program: denaturation at 95 °C for 10 s, annealing at 56 °C for 5 s, and extension at 72 °C for 10 s. Specificity of reaction products was checked by determination of melting points (0.1 °C/s transition rate). All samples were amplified in triplicate, and hypoxanthine phosphoribosyltransferase (*HPRT*) gene was used as a reference to normalize obtained results.

The primers were designed using Primer 3 software (Whitehead Institute for Biomedical Research, Cambridge, MA, USA) (Table 2). They were purchased from the Laboratory of DNA Sequencing and Oligonucleotide Synthesis, Institute of Biochemistry and Biophysics, Polish Academy of Sciences, Warsaw.

### 4.7. Microarray Expression Study

The microarray study was carried out according to the previously described procedure [69,70,71,72]. The previously isolated RNA was pooled into four samples, representing control group (*n* = 2) and experimental group (*n* = 2) where the RNA was isolated after 24 hours from LEP administration (concentration 1 × 10^−6^ M). The protocol involving transcription in vitro, biotin labelling and cDNA fragmentation for further hybridization was carried out using the Affymetrix GeneChip IVT Express Kit (Affymetrix, Santa Clara, CA, USA). Obtained biotin-labelled fragments were hybridized with Affymterix GeneChip Human Genome U219 microarrays together with control cDNA and oligo B2. The hybridization process was conducted with the use of the AccuBlockTM Digital Dry Bath (Labnet International, Inc., Edison, NJ, USA) hybridization oven at 45 °C for 16 h. Then the microarrays were washed and stained according to the technical protocol using the Affymetrix GeneAtlas Fluidics Station (Affymetrix, Santa Clara, CA, USA). The array strips were scanned by the Imaging Station of GeneAtlas System. Preliminary analysis of the scanned chips was performed using Affymetrix GeneAtlasTM Operating Software. The quality of gene expression data was verified using the quality control criteria established by the software. Obtained CEL files were imported into downstream data analysis.

### 4.8. Microarray Data Analysis

All analyses were performed using BioConductor software with the relevant Bioconductor libraries, based on the statistical R programming language. The Robust Multiarray Average (RMA) normalization algorithm implemented in the “Affy” library was used for normalization, background correction, and calculation of the expression values of all of the examined genes [73]. Biological annotation was taken from BioConductor “oligo” package where annotated data frame object was merged with normalized data set, leading to a complete gene data table [74]. Differential expression and statistical assessment were determined by applying the linear models for microarray data implemented in the “limma” library [75]. The selection criteria of a significantly changed gene expression were based on fold difference higher than absolute 2 and *p*-value after false discovery rate (FDR) correction <0.05. The result of such a selection was presented as volcano plot, showing the total number of up- and down-regulated genes. Raw Data files were also deposited in the Gene Expression Omnibus (GEO) repository at the National Center for Biotechnology Information (http://www.ncbi.nlm.nih.gov/geo/) under the GEO accession number GEO: GSE133616.

### 4.9. Assignment of Differentially Expressed Genes to Relevant Gene Ontology (GO) Terms

The whole set of differentially expressed genes (DEGs) were subjected to functional annotation and clusterization using the DAVID (Database for Annotation, Visualization, and Integrated Discovery) bioinformatics tool [76]. Gene symbols of differentially expressed genes were uploaded to DAVID by the “RDAVIDWebService” BioConductor library [77], where DEGs were assigned to relevant GO terms, with subsequent selection of significantly enriched GO terms from GO BP Direct database. The p-values of selected GO terms were corrected using Benjamini-Hochberg correction described as adjusted p-values [78]. Relevant GO ontological groups with adjusted *p*-values below 0.05 and N per group >5 were visualized using bubble plot. Detailed analysis of genes belonging to selected ontological groups, with their expression fold changes, are presented as circos plots using “GOplot” library [79].

### 4.10. Gene Set Enrichment Analysis (GSEA)

Gene Set Enrichment Analysis was used to determine enrichment or depletion in genes expression between two compared biological groups within a priori defined gene sets (GO terms, pathways). The method uses Kolmogorov–Smirnov (K-S) statistical test for identification of significantly enriched or depleted groups of genes [80]. GSEA analysis has been conducted using FGSEA library [81]. Normalised fold change values from all of the genes presented on the microarray were log2 transformed and ordered. Then, a predefined gene sets from Hallmark database (from the Molecular Signatures Database) was selected [82]. Genes belonging to the selected set were ranked according to the difference in their expression level using signal-to-noise ratio with 1000 times permutation. By walking down the ranked list of genes, the enrichment score (ES) was calculated for each selected gene set [83]. ESs were normalized by their gene set size, and false positives were corrected by FDR.

### 4.11. KEGG Signaling Pathways

The Pathview library of the bioconductor was used to generate the p53 and apoptosis signaling pathway [84]. The fold values of significantly changed genes were mapped by colours on native KEGG, p53 (KEGG ID = hsa04115) and apoptosis signaling pathway (KEGG ID = hsa04210), where green represents up-regulated expression and red represents down-regulated expression levels in relation to the control group. In order to show a comprehensive image concerning the regulation of the analysed signaling pathways, all genes whose expression was significantly different without a cut-off at fold values were visualized.

### 4.12. Western Blot Analysis

Cell supernatant was homogenized in RIPA buffer (Sigma Aldrich, St. Louis, MO, USA) with addition of EDTA-free Protease Inhibitor Coctail (Roche, Basel, Switzerland). The concentration of isolated proteins was determined by Bradford Protein Assay (Bio-Rad, CA, USA) [85]. The electrophoresis was conducted on 4–20% Mini Protean TGX Preacast electrophoretic gel (Mini-Protein TGX Bio-Rad) with Tris/Glycine/SDS Buffer (Bio-Rad, CA, USA, no: 1610732). The first gel pocket was filled with markers PageRuler Plus Prestained Protein Ladder (Thermofisher Scientific, Waltham, MA, USA, no 26619). For each sample, 20 µg of protein was separated and transferred onto a PVDF membrane using Trans-Blot Turbo RTA Mini Nitrocellulose Transfer System (Bio-Rad, CA, USA, no: 1704270). Transferred proteins were stained with Ponceau S. Membranes were incubated in a blocking buffer consisting of 5% non-fat dry milk in TBST for 1 h, followed by primary antibody incubation overnight at 4 °C with primary antibody (Cell Signaling Technology, Danvers, Massachusetts, USA): p38 MAPK (cat. no: #8690), p44-42 MAPK (cat. no: #4695), Phospho Bcl-2 (Thr56) (cat. no: #2875), Stat1 (cat. no: #9172), GAPDH (cat. no: sc-47724, Santa Cruz, CA, USA) at dilution 1:1000. Subsequently, membranes were thoroughly washed and incubated with an anti-rabbit IgG HRP-linked antibody in 1:10,000 dilution (#7074; Cell Signaling Technology) for 1 h at room temperature. After washing, membranes were incubated with enhanced chemiluminescence (SuperSignal West Femto Maximum Sensitivity Substrate (#34096 Thermofisher Scientific, Waltham, MA, USA) detection reagents (1 min, room temperature) and visualized on ProteinSimple—Western Blot FluorChem Systems with AlphaView software (ProteinSimple, San Jose, CA, USA). Densitometric analysis was performed in relation to GAPDH.

### 4.13. ELISA Test—MMP7 Level Detection

The culture medium from control and LEP 1 × 10^−6^ M groups was subjected to an analysis of the metalloproteinase 7 (MMP7) secretion level using solid phase enzyme-linked immunosorbent assay (ELISA) test (Abcam, Cambridge, UK, ab100608, MMP7 Human ELISA Kit). All assays were performed according to the manufacturers’ protocols. Absorbance (OD) of each plate wells were measured at 450 nm with Biotek—synergy 2, microtiter plate reader. Quantitative analysis was performed using a Four-Parameter Logistic (4PL) curve, implemented in “drc” package of Bioconductor [86].

### 4.14. Expression of Leptin System in Normal Prostate and Tumor Tissues—Analysis of The Cancer Genome Atlas (TCGA) Dataset

Clinical description with RNAseq data for 52 normal prostate (control) and 498 prostate adenocarcinoma were downloaded from public TCGA database [87] using FireBrowse server (http://gdac.broadinstitute.org/) [88]. Then voom algorithm from “Limma” package was used for data normalization [75]. Normalized data for leptin (*LEP*), leptin receptor (*LEPR*) and its main downstream signaling genes (*JAK2*, *STAT3*), were extracted from the whole dataset. The obtained expression values were subjected to statistical analysis using Mann-Whitney test and visualised as boxplots with medians and interquartile ranges (IQR).

### 4.15. Statistics

Statistical evaluation of the differences between groups was carried out using the Student’s *t*-test or Mann-Whitney test with asterisk annotation (* *p* < 0.05; ** *p* < 0.02; *** *p* < 0.01). Each of the described experiments was repeated at least three times. In the case of data obtained from microarray, differences were evaluated by statistical programs included in particular bioinformatic analyses.

## 5. Conclusions

Obtained results suggest activation of apoptotic processes in LNCaP cells cultured at high LEP concentration. At the same time, this activation is accompanied by inhibition of proliferation of the tested cells.

## Figures and Tables

**Figure 1 ijms-20-05412-f001:**
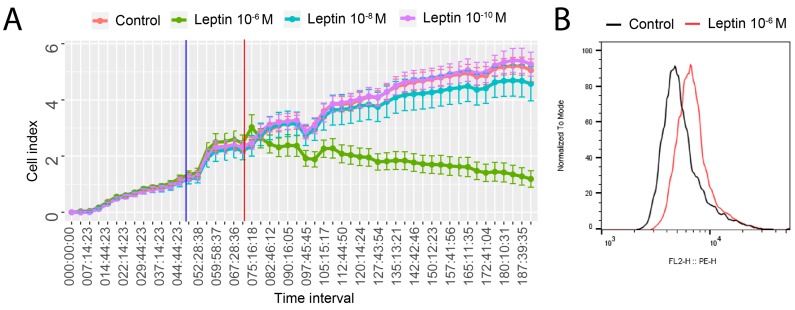
(**A**) Cell index values of LNCaP cells cultivated for 196 h in the presence of various LEP concentrations. Electrical impedance is measured throughout the cultivation period at a 15-minutes frequency. The time point of the medium changed to starvation medium is marked with a blue vertical line. LEP is provided at the time point indicated by a red vertical line. The mean cell index values with SE for the eight repetitions in each group are shown. (**B**) Flow cytometry analysis of Cleaved PARP-1. Median fluorescence intensity, LNCaP cells treated with LEP (1 × 10^−6^ M) in relation to untreated control group.

**Figure 2 ijms-20-05412-f002:**
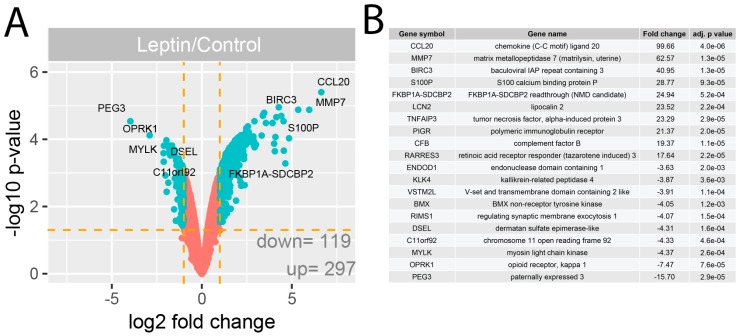
(**A**) Volcano plots of total gene expression profiles of the LNCaP prostate cell line after 24 h incubation with LEP (1 × 10^−6^ M). Each dot represents the mean expression (*n* = 2) of the individual gene obtained from a microarray normalized dataset. The orange dotted lines (cut off values) were established according to the following parameters: fold= |2| and *p*-value with FDR correction = 5%. Genes above the cut-off lines have been considered as differentially expressed genes (DEG) and are shown as turquoise dots. The total numbers of DEG are presented in the bottom right corner of the graph. Ten the most regulated genes are marked by their gene symbols. (**B**) List of 20 genes with the highest (10 genes) and lowest (10 genes) fold changes obtained from the datasets of differentially expressed genes.

**Figure 3 ijms-20-05412-f003:**
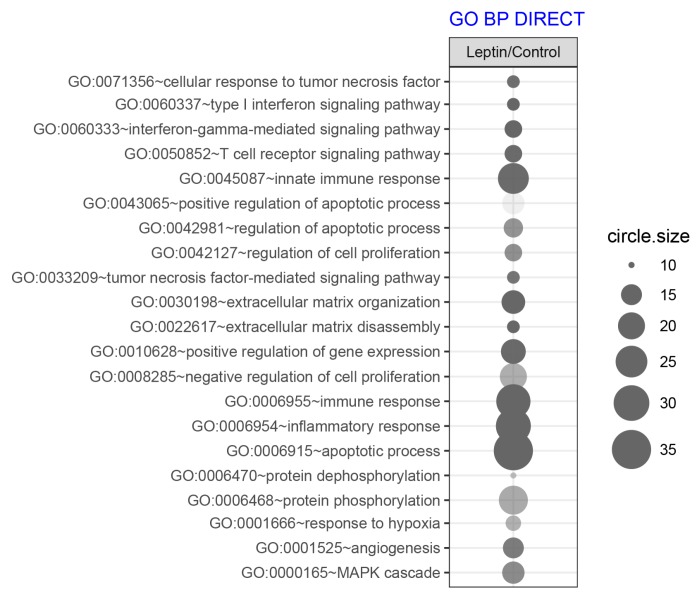
Bubble plot of overrepresented gene sets in DAVID GO BP DIRECT annotations database obtained from comparisons in gene expression profiles between LEP-treated LNCaP vs. control. The graph show only the GO groups above the established cut-off criteria (*p* with correction <0.05, a minimal number of genes per group >10). The size of each bubble reflects the number of differentially expressed genes, assigned to the GO BP terms. Transparency of the bubbles displays *p*-value (more transparent—closer to the border of *p* = 0.05).

**Figure 4 ijms-20-05412-f004:**
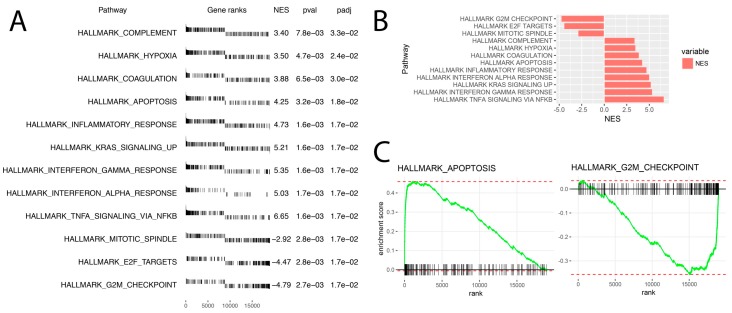
Gene set enrichment analysis using Hallmark gene sets. (**A**) List of significantly enrichment gene sets with appropriate gene ranks, normalized enrichment score (NES), *p* values (pval) and *p* values after FDR correction (padj). (**B**) Bar chart of normalized enrichment score (NES) values for previously selected gene sets. (**C**) Enrichment plot of “Hallmark apoptosis” and “Hallmark G2M checkpoint” gene sets.

**Figure 5 ijms-20-05412-f005:**
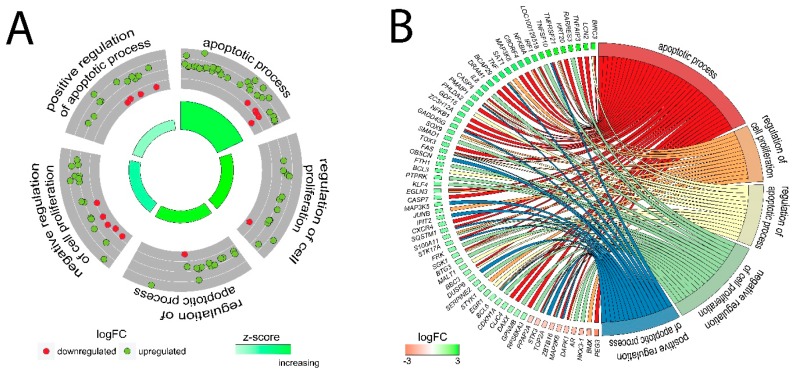
Detailed analysis of five enriched gene ontological groups involved in proliferation/apoptosis. (**A**) The circular scatter plots of differentially expressed genes involved in specific GO terms (positive regulation of apoptotic process, apoptotic process, regulation of cell proliferation, regulation of apoptotic process, negative regulation of cell proliferation). Each dot represents a single gene whose expression is increased (green) or decreased (red) due to LEP action. Positive value of Z-score mapped on a red colour scale was presented inside the graph. (**B**) Circos plots with interdependence between selected GO terms and their genes. Symbols of DEG are presented on the left side of the graph with their fold change values, mapped by colour scale (green = higher expression; red = lower expression). Gene involvement in the GO terms was determined by coloured connecting lines.

**Figure 6 ijms-20-05412-f006:**
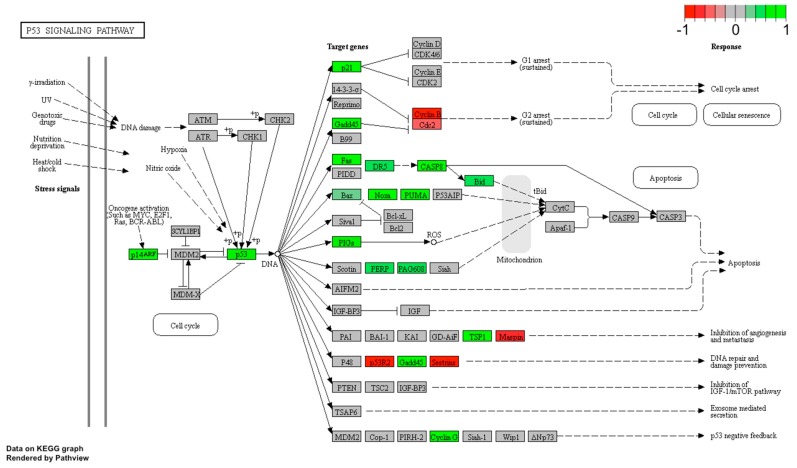
p53 signaling pathway in LNCaP cells treated with LEP (1 × 10^−6^ M). Expression changes of target genes are mapped by colours; green colour—statistically significant increase in expression, red colour—statistically significant decrease in expression, grey colour—statistically insignificant.

**Figure 7 ijms-20-05412-f007:**
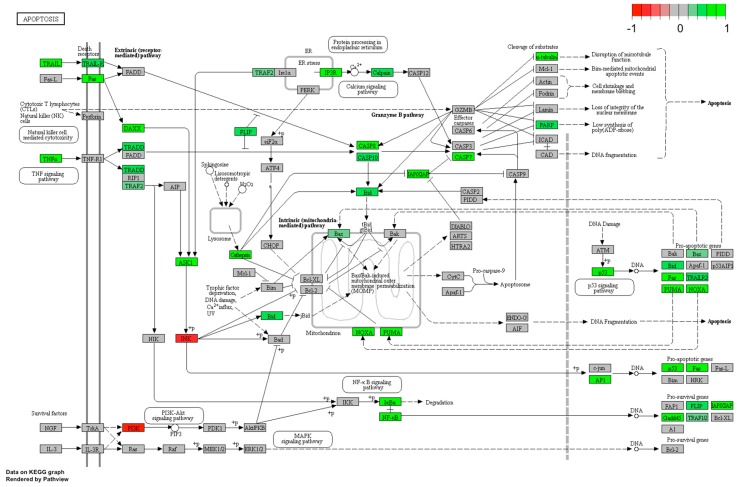
Apoptosis pathway in LNCaP cells treated with LEP (1 × 10^−6^ M). Expression changes of target genes are mapped by colours; green colour—statistically significant increase in expression, red colour—statistically significant decrease in expression, grey colour—expression statistically insignificant.

**Figure 8 ijms-20-05412-f008:**
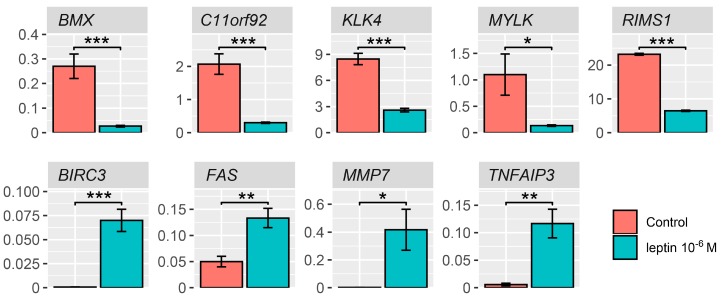
Real time qPCR validation of microarray data. *BMX*—BMX non-receptor tyrosine kinase, *C11orf92*—chromosome 11 open reading frame 92, *KLK4*—kallikrein-related peptidase 4, *MYLK*—myosin light chain kinase, *RIMS1*—regulating synaptic membrane exocytosis 1, *BIRC3*—baculoviral IAP repeat containing, *FAS*—apoptosis signal receptor/cell surface death receptor, *MMP7*—matrix metallopeptidase 7, *TNFAIP3*—tumor necrosis factor, alpha-induced protein 3. Results are presented as means ± SEM, *n* = 3. Statistical comparison by Student’s t-test: * *p* < 0.05; ** *p* < 0.02; *** *p* < 0.01.

**Figure 9 ijms-20-05412-f009:**
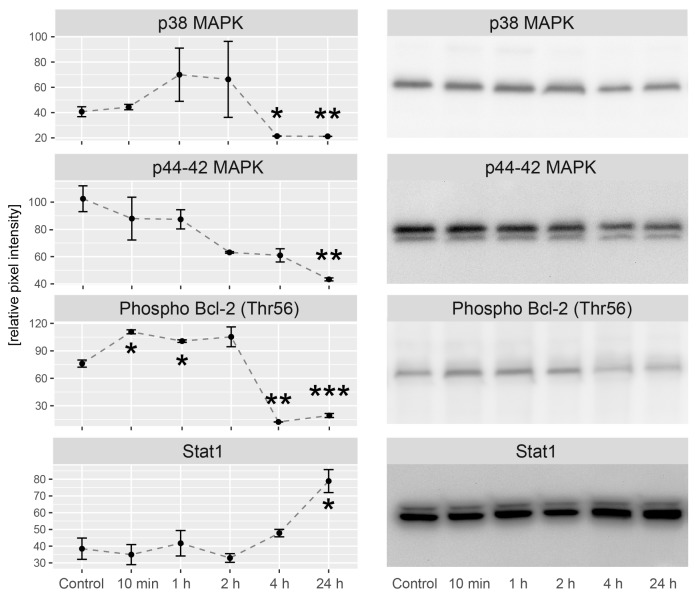
Densitometric analysis includes several key factors of signaling pathways involved in apoptosis, proliferation in different time after the treatment with LEP. Graphs represent protein expression relative to glyceraldehyde-3-phosphate dehydrogenase (GAPDH) levels. Representative experiments of proteins immunoblotting were also shown. Results are presented as means ± SEM, *n* = 3. Statistical comparison by Student’s t-test: * *p* <0.05, ** *p*<0.01, *** *p*<0.001.

**Figure 10 ijms-20-05412-f010:**
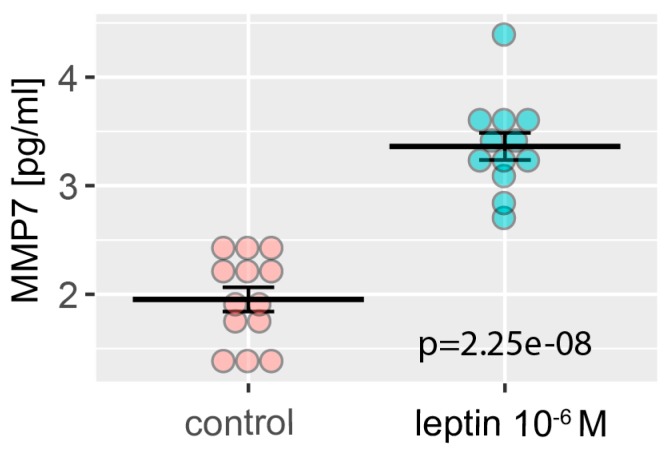
Level of MMP7 in LNCaP cells medium with (blue) or without (red) LEP at 1 × 10^−6^ M concentration. Levels of MMP7 were measured by ELISA, *n* = 12 (10^−6^ M) and *n* = 13 (control). Results are presented as means ± SEM. Each dot represents an individual sample. Statistical comparison by Student’s t-test: *p* value shown on the graph.

**Figure 11 ijms-20-05412-f011:**
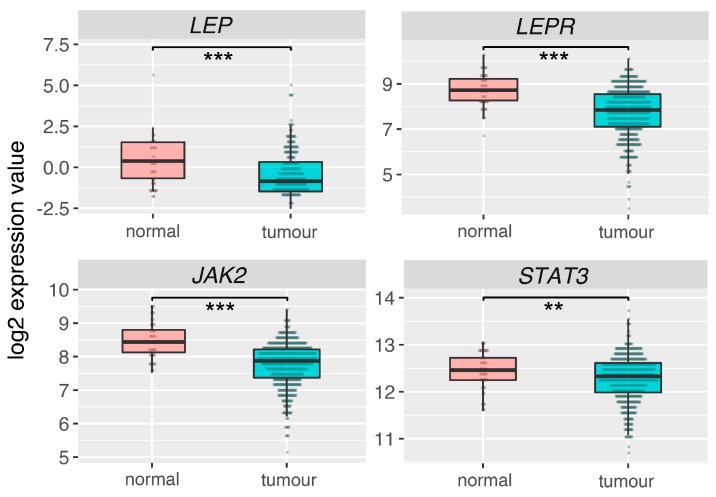
Expression of leptin (*LEP*), leptin receptor (*LEPR*) and its main downstream signaling genes (*JAK2*, *STAT3*) in normal prostate and adenocarcinoma. Analysis based on data from The Cancer Genome Atlas (TCGA). Results are presented as medians with IQRs. Statistical comparison by Mann-Whitney test: ** *p* < 0.02; *** *p* < 0.01.

**Table 1 ijms-20-05412-t001:** Literature data on the effect of leptin (LEP) on proliferation and apoptosis in human normal prostate and prostate cancer cell lines.

Cell Type	Method of Detection	Dose of LEP and Time	Effect	Reference
LNCaP-FGC, DU145, PC-3	[3H]thymidine incorporation MTT assay	LEP (0.1–12.5 µg/mL) for 20 h LEP (12.5 µg/mL) for 5 days	stimulation of proliferation in DU145 and PC-3 cells but not in LNCaP-FGC cell	Onuma et al. (2003) [25]
DU145 and PC-3	MTT assay	LEP (0.4 ng/mL and 4.0 ng/mL) for 24 and 48 h	in both cell lines LEP inhibited growth	Somasundar et al. (2003) [52]
DU145 and PC-3	MTT assay ELISA	LEP (4 or 40 ng/mL) for 24 and 72 h LEP (4 or 40 ng/mL) for 24 h	stimulation of proliferation in both cell lines increased apoptosis in both cell lines	Somasundar et al. (2004) [53]
DU145	[3H]thymidine incorporation	LEP (12.5 μg/mL) for 20 h	stimulation of proliferation	Miyazaki et al. (2008) [29]
LNCaP, DU-145 and PC-3	[3H]thymidine incorporation	LEP (20–200 ng/mL) for 144h	stimulation of proliferation of LNCaP cells, no effects on PC-3 and Du-145 cells	Deo et al. (2008) [27]
DU145 and PC-3	XTT colorimetric assay (tetrazolium based assay) Cell death detection ELISA plus^®^ assay	LEP (5–100 ng/mL) for up to 48 h LEP (100 ng/mL) for 24 h	stimulation of proliferation in both cell lines anti-apoptotic effects	Hoda & Popken (2008) [54]
LNCaP and PC3	CellTiter 96^®^ AQueous One Solution Cell Proliferation Assay (tetrazolium based assay)	LEP (0.01–100 nM) for 48 h	proliferation in LNCaP cells unaffected, in PC3 cells significantly increased	Mistry et al. (2008) [26]
LNCaP-FGC, DU-145, PC-3, and PC-3 cells stably expressing AR (androgen receptor)—PC-3/AR	apoptosis effector protein caspase 3 levels, cleavage of the DNA repair enzyme, and the numbers of apoptotic cells visualized by Hoechst 33342	LEP (1 ng/mL) for 24 h	In all studied cell lines statistically significant pro-apoptotic effects of LEP	Samuel-Mendelsohn et al. (2011) [35]
samples of human hyperplastic prostate tissue	cell proliferation evaluated by immunohistochemistry for PCNA, RT-PCR (expression of apoptosis related genes)	incubation with LEP (50 ng/mL) for 3 h	stimulation of cell proliferation (ca 3-folds) and BAX expression, lowered expression levels of BCL-2 and BCL-X	Leze et al. (2012) [23]
LNCaP, DU145 and PC-3	WST-8 assays and a Cell Counting kit-8	LEP (up to 1000 ng/mL) for 48 h	in all cell lines no effects on cell proliferation, LEP (100 ng/mL) - cell number notably increased between days 7–42 of culture	Noda et al. (2015) [28]
human normal prostate (PrEC, PrSC, PrSMC) and prostate cancer (DU145, LNCaP, PC3) cell lines	proliferative activity was determined by RTCA (real-time cell analyzer)	LEP (1 × 10^−6^, 1 × 10^−8^ and 1 × 10^−10^ M) for at least 70 h	lowered proliferation rate of LNCaP cells - at 1 × 10^−6^ M LEP concentration, increased proliferation rate of DU145 cells at the same concentration, increased proliferation rate of PrSC cells at 1 × 10^−8^ and 1 × 10^−10^ M LEP concentrations. In all remaining tests LEP did not influence the proliferation rate of the studied cells	Szyszka et al. (2018) [24]

**Table 2 ijms-20-05412-t002:** Oligonucleotide sequences of sense (S) and antisense (A) primers.

Gene Symbol	Genbank Accession Number	Primer	Primer Sequence (5′–3′)	Position	PCR Product Size (bp)
*BMX*	NM_203281.3	S	CATCGGACACCATCTACCAG	2053–2072	278
	BMX non-receptor tyrosine kinase	A	CTTTTGTTTCCTGCCTTGTTC	2310–2330	
*C11orf92*	NM_001302644.1	S	AGCAAGAATATCACCGTGAAGCA	129–151	184
	chromosome 11 open reading frame 92	A	ACCACGATGTCGGGTAACTC	293–312	
*KLK4*	NM_004917.4	S	CTCTATGACCCGCTGTACCAC	541–561	123
	kallikrein-related peptidase 4	A	CACAAGGCCCTGCAAGTACC	644–663	
*MYLK*	NM_053025.4	S	AAGTGCTGCTAGATTTGACT	5781–5800	136
	myosin light chain kinase	A	AATTAAAGAGCAGTTCCCGTC	5896–5916	
*RIMS1*	NM_014989.5	S	AATATTTCCTGGAGTGCGACTGG	4830–4852	179
	regulating synaptic membrane exocytosis 1	A	GGCTTCGTGCTCTAATGACTT	4988–5008	
*MMP7*	NM_002423.4	S	ATGATATTAAAGGCATTCAGA	799–819	281
	matrix metallopeptidase 7	A	TTTATTGACATCTACCCACT	1060–1079	
*TNF IP3*	NM_001270508.1	S	CCATCATTTTGTACCCTTG	1132–1150	280
	TNF alpha induced protein 3 transcript variant 1	A	TTCAAGTAATCATCTACCAG	1391–1411	
*FAS*	NM_000043.5	S	CAAAAGTGTTAATGCCCAA	378–396	299
	Fas cell surface death receptor transcript variant 1	A	TGCAGTTTATTTCCACTTC	658–676	
*BIRC3*	NM_001165.4	S	TAGTAAAAGGAAATATTGCAG	4294–4314	100
	baculoviral IAP repeat containing 3 transcript variant 1	A	TATTTTATGTCCTGTTGCAC	4374–4393	
*HPRT*	NM_000194.2	S	GCCATCACATTGTAGCCCTC	343–362	172
	hypoxanthine phosphoribosyltransferase 1	A	ACTTTTATGTCCCCTGTTGACT	493–514	

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
