# Peer review of "Analysis of Transcriptome, Selected Intracellular Signaling Pathways, Proliferation and Apoptosis of LNCaP Cells Exposed to High Leptin Concentrations"

_ijms, 2019, doi:10.3390/ijms20215412_

Round 1

Reviewer 1 Report

Ad 1. The added paragraph is insufficient, as it does not inform the reader about key leptin roles in processes other than energy balance and food intake. It is absolutely crucial because levels of leptin used in the experiments were extremely high when compared to its physiological and even pathological, e.g. associated with obesity) concentrations. Theoretically speaking, if leptin is to be used in the treatment of prostate cancer, such excessive levels could negatively affect these processes. Also, the new fragment doubles the information regarding cancer already present in the Introduction section. Therefore, with the new fragment, the sentence “Numerous population studies suggest that obesity is a risk factor associated with the development of various types of cancer,…” is not necessary, as “various types of cancer” have been listed in the preceding, newly added sentences.

Ad. 2a. OK.

Ad. 2b. OK.

Ad. 3. I think that this fragment of Discussion should benefit from additional sentence/sentences on the potential use of leptin in the treatment of prostate cancer, provided that the hormone is delivered directly to cancer to avoid adverse systemic effects. This could increase the validity of results.

Ad. Minor point: I probably did not make myself clear. I think that the Authors should pay more attention to the biological relevance of their results. Indeed, at the beginning of Discussion, they provide information regarding molecular pathways preferentially activated by the long and short versions of the leptin receptors. However, the information that is missing regards the BIOLOGICAL processes that could be differentially activated/inhibited when different receptor variants are used. Use of various receptor isoforms by a single hormone to produce different/opposite biological effects is a widespread mechanism of action of hormones.   

Author Response

Prostate cancer is common. If the disease is limited to this organ or only locally infiltrating, the 5-year relative survival rate is nearly 100%. However, when distant metastases are present, the survival rate decreases to only 30%. Moreover, advanced prostate cancers become resistant to hormonal therapies and, therefore, there is an urgent need to find another method of treatment of such cases.

In their article, the Authors describe the effects of a very high concentration of leptin on the function of the androgen-sensitive LNCaP line of prostate cancer cells. This cell line is commonly used in studies on the pathophysiology of this cancer and, in the absence of tissue explants or primary cultures, its use seems to be justified. On the other hand, the normal concentration of leptin in men should not exceed ten ng/ml, while anti-proliferative and pro-apoptotic effects were observed at levels exceeding this value a few hundred (!) times and concentrations relatively closed to physiological ones were ineffective.

Upon analysis of this manuscript, I found a few major points to be addressed by the Authors:

Q 1.1. The introduction should include a paragraph describing the physiological roles of leptin and its adverse roles at high levels (hyperleptinemia) at the level of the whole organism and the molecular level.

Answer1.1: First of all, we would like to thank You for your insightful remarks and efforts reviewing our publication.I agree that information concerning the physiological role of leptin and hyperleptinemia will constitute a valuable justification for our research. In accordance with the Reviewer's suggestions, such information was added to the introduction section.

Q 1.2: The added paragraph is insufficient, as it does not inform the reader about key leptin roles in processes other than energy balance and food intake. It is absolutely crucial because levels of leptin used in the experiments were extremely high when compared to its physiological and even pathological, e.g. associated with obesity) concentrations. Theoretically speaking, if leptin is to be used in the treatment of prostate cancer, such excessive levels could negatively affect these processes. Also, the new fragment doubles the information regarding cancer already present in the Introduction section. Therefore, with the new fragment, the sentence “Numerous population studies suggest that obesity is a risk factor associated with the development of various types of cancer,…” is not necessary, as “various types of cancer” have been listed in the preceding, newly added sentences.

Answer1.1: Once again I would like to thank You for the accurate and substantive review of my publication. The introduction was improved in accordance with the reviewer's suggestions.

Q2.1 - Discussion: is too extensive and sometimes too detailed. On the other hand, essential parts seem to be missing. For example, although the Authors mentioned that they were aware that the leptin concentration used in their experiments was very high, only extremely briefly they mentioned that others sometimes used such concentrations in in vitro studies, and did not discuss this most important matter anymore. Specifically, they did not relate it to any pathophysiological condition, did not discuss whether there is any chance to use such concentration in treatment protocols. In other words – there is no conclusion whether their results are of any value in the real world.

Answer 2.1: Thank you for that remark. Unfortunately, the discussion in transcriptomic studies is problematic because it is impossible to discuss all the obtained results, however the relevant paragraph has been added to the discussion. The practical importance of the study was also supplemented by the analysis of RNAseq data (extracted from TCGA) for 52 normal prostate (control) and 498 prostate adenocarcinoma, where we found a decrease in expression of leptin (LEP), leptin receptor (LEPR) and its main downstream signaling genes (JAK2, STAT3).

Q 2.2 I think that this fragment of Discussion should benefit from additional sentence/sentences on the potential use of leptin in the treatment of prostate cancer, provided that the hormone is delivered directly to cancer to avoid adverse systemic effects. This could increase the validity of results.

Answer 2.1: Relevant sentences have been added to the discussion.

Minor point 1.1:

I think it should be emphasized more clearly that the observed effects of leptin in the LNCaP cells occur via type 2 leptin receptor. A small paragraph discussing the different effects of activation of different receptor types would be interesting.

Answer Minor point 1.1: This aspect was described at the beginning of discussion and in our previous paper: Szyszka, M.; Tyczewska, M.; Milecka, P.; Jopek, K.; Celichowski, P.; Malendowicz, L. K.; Rucinski, M., Effects of leptin on leptin receptor isoform expression and proliferative activity in human normal prostate and prostate cancer cell lines. Oncol Rep 2018, 39, (1), 182-192.

Minor point 1.2: I probably did not make myself clear. I think that the Authors should pay more attention to the biological relevance of their results. Indeed, at the beginning of Discussion, they provide information regarding molecular pathways preferentially activated by the long and short versions of the leptin receptors. However, the information that is missing regards the BIOLOGICAL processes that could be differentially activated/inhibited when different receptor variants are used. Use of various receptor isoforms by a single hormone to produce different/opposite biological effects is a widespread mechanism of action of hormones.

Answer Minor point 1.2:

Unfortunately, data concerning the biological role of leptin receptors mainly focused on LEPR var. 1 (Obr-b). This receptor is highly expressed in the hypothalamus, where it plays a role in the main neuroendocrine effects of leptin causes reduced food intake and increased energy expenditure. Data on the physiological role of other receptors are limited. A few papers focused on the role of short LEPR (obr a and obr c) in the translocation of leptin through cell barriers, such as the blood-brain barrier and the placenta. I agree that recognising the role of each LEPR variant would provide a better explanation of the often contradictory leptin effects, but with the best of my knowledge, there are currently no such reports.

Reviewer 2 Report

I read the revised version of the manuscript.
My recommendation is that it is acceptable as it is.

Author Response

Thank you so much for accepting my manuscript.

Best regards.

This manuscript is a resubmission of an earlier submission. The following is a list of the peer review reports and author responses from that submission.

Round 1

Reviewer 1 Report

Prostate cancer is common. If the disease is limited to this organ or only locally infiltrating, the 5-year relative survival rate is nearly 100%. However, when distant metastases are present, the survival rate decreases to only 30%. Moreover, advanced prostate cancers become resistant to hormonal therapies and, therefore, there is an urgent need to find another method of treatment of such cases.

In their article, the Authors describe the effects of a very high concentration of leptin on the function of the androgen-sensitive LNCaP line of prostate cancer cells. This cell line is commonly used in studies on the pathophysiology of this cancer and, in the absence of tissue explants or primary cultures, its use seems to be justified. On the other hand, the normal concentration of leptin in men should not exceed ten ng/ml, while anti-proliferative and pro-apoptotic effects were observed at levels exceeding this value a few hundred (!) times and concentrations relatively closed to physiological ones were ineffective.

Upon analysis of this manuscript, I found a few major points to be addressed by the Authors:
1. The introduction should include a paragraph describing the physiological roles of leptin and its adverse roles at high levels (hyperleptinemia) at the level of the whole organism and the molecular level.
2. Results: in silico analysis of data is extensive and seems to be correct. However, validation data should be widened:
a. Real-time qPCR was used only to validate data regarding leptin-activated genes. Data regarding leptin-inhibited genes are missing.
b. Immunoblot validation of protein levels: major cellular processes affected by leptin were proliferation and apoptosis. This analysis should include proteins involved in these processes, encoded by genes mRNAs of which were affected by the leptin treatment.
3. Discussion: is too extensive and sometimes too detailed. On the other hand, essential parts seem to be missing. For example, although the Authors mentioned that they were aware that the leptin concentration used in their experiments was very high, only extremely briefly they mentioned that others sometimes used such concentrations in in vitro studies, and did not discuss this most important matter anymore. Specifically, they did not relate it to any pathophysiological condition, did not discuss whether there is any chance to use such concentration in treatment protocols. In other words – there is no conclusion whether their results are of any value in the real world.

Minor point:
I think it should be emphasized more clearly that the observed effects of leptin in the LNCaP cells occur via type 2 leptin receptor. A small paragraph discussing the different effects of activation of different receptor types would be interesting.

Reviewer 2 Report

The manuscript by Szyszka et al described their work on the downstream signals induced by high dose of leptin treatment in human prostate cancer cell line, LNCaP cells. They performed microarray analysis to identify differentially expressed genes. Then they found that leptin has effects on apoptosis-related gene expression or the secretion of matrix metallopeptidase 7. Thus, they demonstrated the anti-proliferative effect of high dose of leptin in LNCaP cells by regulating these signals.

While this study presented some novel evidences to show the effect of leptin treatment in prostate cancer cells, there are several weaknesses described below in the current manuscript that needs to be strengthened.

Major points

In this manuscript, clinical relevance of leptin is not mentioned. They should describe how leptin concentration is controlled or measured in prostate cancer patients in the past reports. In addition, it is desirable to show how the downstream signals of leptin are associated with clinical prostate cancer progression by analyzing expression of these signals in clinical samples. In their microarray study, they identified apoptosis-related pathway is the most significantly regulated by leptin treatment (P4, Ln.125). However, the authors have not evaluated the effect of leptin on the induction of apoptosis. Although they showed a result of cell proliferation assay in Figure 1, it is not sufficient and more functional assay such as TUNEL assay is necessary to demonstrate that the apoptosis related signals are regulated by leptin.

Minor

In their western blot analysis (Figure 9), expression level of loading control or non-phosphorylated protein (such as total MAPK) are not shown. We could not determine how these phosphorylation events are controlled by these results. In Figure 5A, the authors showed the number of genes in “apoptosis process”, “regulation of apoptosis process”, and “positive regulation of apoptotic process”. What is the difference between these three Go-terms? Moreover, they showed “regulation of cell proliferation” and “negative regulation of cell proliferation”. Do these terms mean “positive” and “negative” regulation of cell proliferation respectively?